# Automated app-based augmented reality cognitive behavioral therapy for spider phobia: Study protocol for a randomized controlled trial

**Marieke B. J. Toffolo** [1,2], **Jamie R. Fehribach**[1,2], **Chris P. B. J. van Klaveren**[3,4], **Ilja Cornelisz**[3,4], **Annemieke van Straten**[1,2], **Jean-Louis van Gelder**[5,6], **Tara Donker**[1,2,7] *

1 Department of Clinical, Neuro-, and Developmental Psychology, Section Clinical Psychology, Vrije Universiteit Amsterdam, Amsterdam, The Netherlands, 2 Amsterdam Public Health Research Institute Amsterdam, Amsterdam, The Netherlands, 3 Department of Education Sciences, Section Methods and Statistics, Vrije Universiteit Amsterdam, Amsterdam, The Netherlands, 4 Amsterdam Center for Learning Analytics (ACLA), Amsterdam, The Netherlands, 5 Max Planck Institute for the Study of Crime, Security and Law, Freiburg im Breisgau, Germany, 6 Institute of Education and Child Studies, Leiden University, Leiden, The Netherlands, 7 Department of Psychology, Laboratory of Biological and Personality Psychology, Albert Ludwigs-University of Freiburg, Freiburg im Breisgau, Germany

* t.donker@vu.nl

**Data Availability Statement:** No datasets were generated or analysed during the current study. All

## Abstract

### Background

Fear of spiders, or Arachnophobia, is one of the most common specific phobias. The gold standard treatment, *in vivo* exposure therapy, is effective, but comes with significant limitations, including restricted availability, high costs, and high refusal rates. Novel technologies, such as augmented reality, may help to overcome these limitations and make Exposure Therapy more accessible by using mobile devices.

### Objective

This study will use a Randomized Controlled Trial design to investigate whether ZeroPhobia: Arachnophobia, a 6-week Augmented Reality Exposure Therapy smartphone *self-help* application, can effectively reduce spider phobia symptoms. Additionally, we will examine user-friendliness of the application and the effect of usage intensity and presence on treatment outcome.

### Methods

This study is registered in the Netherlands Trial Registry under NL70238.029.19 (Trial NL9221). Ethical approval was received on October 11, 2019. One-hundred-twelve participants (age 18–64, score $\geq$ 59) on the Fear of Spiders Questionnaire [FSQ] will be recruited from the general Dutch population and randomly assigned to a treatment or waitlist control group. The ZeroPhobia application can be accessed on users' smartphone. Baseline, post-test (i.e., at six weeks), 3- and 12-month follow-up assessments will be done, each including the Fear of Spiders Questionnaire as the main outcome measure as well as additional

relevant data from this study will be made available upon study completion.

**Funding:** This study has been funded by the Dutch Research Council (NWO) Aspasia grant (015.014.072), NWO Creative Industrie-KIEM (KI.18.039), and an MIT R&D grant from Province of North Holland, the Netherlands. All funding was awarded to TD. The funders had and will not have a role in study design, data collection and analysis, decision to publish, or preparation of the manuscript.

**Competing interests:** I have read the journal's policy and the authors of this manuscript have the following competing interests: TD and JLvG have developed the ZeroPhobia application, which is used in the present study in collaboration with Vrije Universiteit. ZeroPhobia is intended for commercial release. Hence, TD and JLvG will not be involved in data analysis or with any decisions related to the publication of findings. This does not alter our adherence to PLOS ONE policies on sharing data and materials. The other authors have declared that no competing interests exist.

measures of anxiety, depression, user-friendliness, and presence as secondary measures and covariates.

## Results

The study was funded on September 25, 2018. Data collection started in September 2021 and the study is expected to run until September 2022.

## Conclusions

Our study will improve our understanding of the efficacy and feasibility of providing Exposure Therapy for spider phobia using an Augmented Reality self-help application, with the intention of making mental health care more accessible.

## Introduction

With an average lifetime prevalence of 7.2%, specific phobia are one of the most common anxiety disorders world-wide [1]. They are characterized by an exaggerated, irrational fear of a specific object or situation that substantially interferes with the person's ability to function [2]. The object or situation almost always evokes an immediate fear response and is therefore either actively avoided or endured with intense anxiety. Fear of animals, including spiders, is one of the most common subtypes of specific phobia [1]. Moreover, fear of spiders (or arachnophobia) can be particularly impairing because spiders can be encountered unexpectedly in common areas in and around the house or outdoors. Additionally, spiders are often viewed as unpredictable because of unique locomotion patterns where their bodies remain relatively stationary while their legs move at a fast and seemingly unsynchronized pace [3]. Thus, arachnophobia is not only highly impairing because of how common spiders are, but also because these movement-related characteristics make them particularly anxiety provoking.

The treatment of choice for specific phobia in general, including arachnophobia, is exposure therapy (ET; e.g., [4]). This treatment consists of gradually exposing the individual to the feared object or situation to allow them to gain experiences that disconfirm their fear expectancies and create new adaptive learning [5]. ET most commonly takes place with a therapist in real world situations (*in vivo*). However, this set-up not only comes with limited availability and high costs, but also with high patient refusal rates due to fear of the procedure [6]. Therefore, novel technologies, such as virtual reality (VR) and augmented reality (AR), have been investigated to overcome these limitations and make treatment more accessible and acceptable [7]. In virtual reality exposure therapy (VRET), artificially created, computer-generated environments replace *in vivo* therapy in order to expose clients to their phobia. Extensive research has shown that VRET is as effective as traditional forms of exposure therapy for treatment of anxiety disorders in general, and for specific phobias in particular [e.g., 8–12]. Moreover, VRET was found to be more acceptable and to have a lower refusal rate than *in vivo* exposure [13]. Mobile VRET applications, which provide more flexible, automated, gamified therapeutic methods are also effective for treatment of specific phobia [14–16]. However, evidence on the feasibility and efficacy of AR exposure therapy (ARET) is limited.

AR technology places virtual content into the real world, combining real and virtual objects in real-time and thus enhancing ("augmenting") the user's sensory perception of reality [17]. Unlike VR, the user is not immersed in an entire virtual environment. Instead, an AR system takes in the real-time surrounding environment via a camera and a virtual element is

superimposed on this image. AR can be experienced using head mounted displays (HMD), but also through handheld devices, such as smartphones or tablets [e.g., 18]. The latter makes AR technology more easily accessible than VR for most people, because the majority of the population has access to this kind of technology (smartphone penetration rate within Western Europe and the U.S. is $\geq$ 75%; [19]). Additionally, when using a handheld device there is no need for an additional headset as is the case with VR, which makes the use of AR less costly. Moreover, ecological validity is arguably higher in AR, because the feared stimulus is embedded in the real environment of the individual. Finally, AR has the same advantages as VR in psychological treatment, such as total control over the execution of exposures, easy access to threatening stimuli, possibility of using multiple scenarios, and no risk of real danger to the patient.

Studies on augmented reality exposure therapy (ARET) for small animal phobia show promising results. The first case studies and open trials that used ARET for treatment of cockroach phobia and spider phobia by the use of HMDs showed that the AR environments produced significant anxiety and participants were able to immerse themselves in these environments [20–22]. Subsequently, ARET effectively reduced the patient's fear and avoidance of their target animal and, importantly, participants were able to approach and interact with live cockroaches and spiders. Moreover, treatment gains were maintained at 3-, 6- and 12-month follow-up [22]. However, these initial studies used small sample sizes and did not include a control group.

One of the first studies to utilize a mobile phone game to enhance exposure therapy in combination with AR for treatment of specific phobia was by Botella and colleagues [23]. They showed that this set-up was also able to elicit anxiety and produce decreases in levels of fear, avoidance, and beliefs in catastrophic thoughts, and this progress was maintained up to a one-year follow-up. However, this study only used a single case design and AR was still administered by the use of a HMD and computer.

The first larger randomized control trial [24] compared ARET (administered with the use of a HMD and computer) with in vivo ET for treatment of small animal phobia and indicated that both treatments produced significant reductions in fear and avoidance, which were maintained at follow-up. Additionally, ARET was well-accepted by participants and rated as less aversive than *in vivo* ET. Furthermore, a recent analysis on combined data found that VRET, ARET and *in vivo* ET were all equally effective treatments for small animal phobia [25]. However, the use of HMDs in these studies limits accessibility in real-world applications because of its size and cost.

A recent, much larger study further tested the potential of an AR smartphone application to elicit fear of small animals [18]. In two experiments using a general population sample consisting of anxious, but not phobic, individuals, they showed increasing levels of self-reported anxiety with increasing levels of exposure difficulty. However, individuals who reported low levels of perceived realism also experienced less anxiety. Overall, moving closer to the virtual animal and virtual animals in locomotion elicited the most anxiety. Thus, mobile devices using AR to present virtual animals can evoke anxiety, which is a pre-requisite for ARET implementation. However, De Witte and colleagues [18] did not test the effectiveness of this app as an intervention for small animal fears, but only the level of anxiety that could be induced. Only one, very recent, study tested the effectiveness of a smartphone-based, AR exposure application for treatment of spider fear in a natural, real-life setting [26]. Participants were asked to train with the application for at least 30 minutes per week during a total period of six weeks. This was done in their own environment and at their own convenience. Compared to a no-intervention control group, participants in the intervention group showed significantly lower levels of subjective fear in a Behavioral Approach Task (BAT) with a real spider and on self-report questionnaires of spider fear after treatment, at medium to large effect sizes. Although these results are promising for the use of self-help applications for spider phobia, the study used a

relatively small sample size, there was no long-term follow-up, and participants still attended two full research days where they were in contact with the research team. Therefore, as far as we know, no research to date has been conducted in a natural, real-life setting with a fully automated self-help application.

Therefore, we developed ZeroPhobia: Arachnophobia, an ARET smartphone self-help application for treating spider phobia. This treatment is not guided by an actual therapist and only requires the user to have a compatible but standard smartphone. A previous version of ZeroPhobia, using VRET for treatment of fear of heights, found that the application improved specific phobia symptoms compared with a waitlist control group, with an effect size of $d = 1.14$ (Intention-to-treat [14]).

Thus, although the use of gamified VRET for the treatment of specific phobia is well established, the proposed study will be the first to investigate a mobile, unguided, ARET application for this population in a real life setting where participants will never be in physical contact with a therapist or research team. The main aim of this randomized controlled trial is to investigate whether the AR self-help application ZeroPhobia: Arachnophobia is effective in reducing spider phobia symptoms at post-test compared to a waitlist control group, and whether this effect is maintained at 12-month follow-up. Furthermore, we will test the effects of the application on anxiety and depressive symptoms. Lastly, we will investigate whether ZeroPhobia: Arachnophobia is rated as user-friendly, how intensely it was used, and whether participants felt present and immersed in AR.

We expect that, compared with a waitlist control group, participants in the treatment condition (receiving ZeroPhobia: Arachnophobia) will demonstrate a greater reduction in spider anxiety, general anxiety and depressive symptoms from pre- to post-intervention, and that this will be maintained at 3- and 12-month follow-ups. Additionally, we expect ZeroPhobia: Arachnophobia to be evaluated as user-friendly and that participants will experience presence in the AR environments. Lastly, we hypothesize that a greater reduction in spider anxiety symptoms at post-test will be related to higher usage intensity, sense of perceived realism, and perceived user-friendliness.

## Methods

### Study design

The present study is a randomized controlled trial (RCT) comparing two conditions: intervention and waitlist control. The intervention consists of a self-help smartphone application that participants will follow at their own pace and in their own private environment. The intervention takes six weeks to complete, after which the waitlist control group receives access to the intervention. A total of 112 participants ($n = 56$ per condition) will be recruited (see "sample size" below). All participants will complete online questionnaires at baseline and post-test. Additionally, participants in the intervention group will complete questionnaires at 3- and 12-month follow-up. Participants in the waitlist control group will complete the primary outcome measure six weeks after the post-test (see Fig 1 for SPIRIT schedule of enrollment, interventions, and assessments and see Fig 2 for participant flow chart).

Ethical approval was received from the Medical Ethical Committee of Vrije Universiteit Medical Center (registration number: 2019–321). This study is registered in the Netherlands Trial Registry (NTR) under NL70238.029.19 (Trial NL9221).

### Inclusion and exclusion criteria

To be included, participants must score 59 or above (based on [27]) on the Dutch version of the Fear of Spiders Questionnaire (FSQ; [28]). Additionally, inclusion criteria consist of being between the ages of 18 and 64, being proficient in Dutch, possessing a compatible smartphone

| | STUDY PERIOD | | | | | |
|---|---|---|---|---|---|---|
| | Enrolment | Allocation | Test Phase | Close-out | | |
| TIMEPOINT | $-t_1$ | 0 | 6 weeks | $t_1$ | $t_2$ | $t_3$ |
| **ENROLMENT:** | | | | | | |
| **Eligibility screen** | X | | | | | |
| **Informed consent** | X | | | | | |
| *Baseline* | | X | | | | |
| **Allocation** | | X | | | | |
| **INTERVENTIONS:** | | | | | | |
| *Treatment* | | | X | X | X | X |
| *Waitlist Control* | | | X | X | | |
| **ASSESSMENTS:** | | | | | | |
| *FSQ* | X | | | X | X | X |
| *SPQ* | | X | | X | X | X |
| *BAI* | | X | | X | X | X |
| *PHQ-9* | | X | | X | X | X |
| *IRI* | | X | | | | |
| *Specific Spider exposure questions* | | X | | | | |
| *CEQ* | | X | | | | |
| *CSQ* | | | | X* | | |
| *IPQ* | | | | X* | | |
| *SUS* | | | | X* | | |
| *EMA* | | | X | | | |

* only given to the experimental group

**Fig 1. Standard Protocol Items: Recommendations for Interventional Trials (SPIRIT) schedule of enrollment, interventions, and assessments.**

(either iPhone 6s or higher with iOS 12 and up or Android smartphones operating Nougat 7.0 or newer Android smartphones supporting ARCore, with access to the internet and a Dutch phone number), living in the Netherlands, and having provided written informed consent to

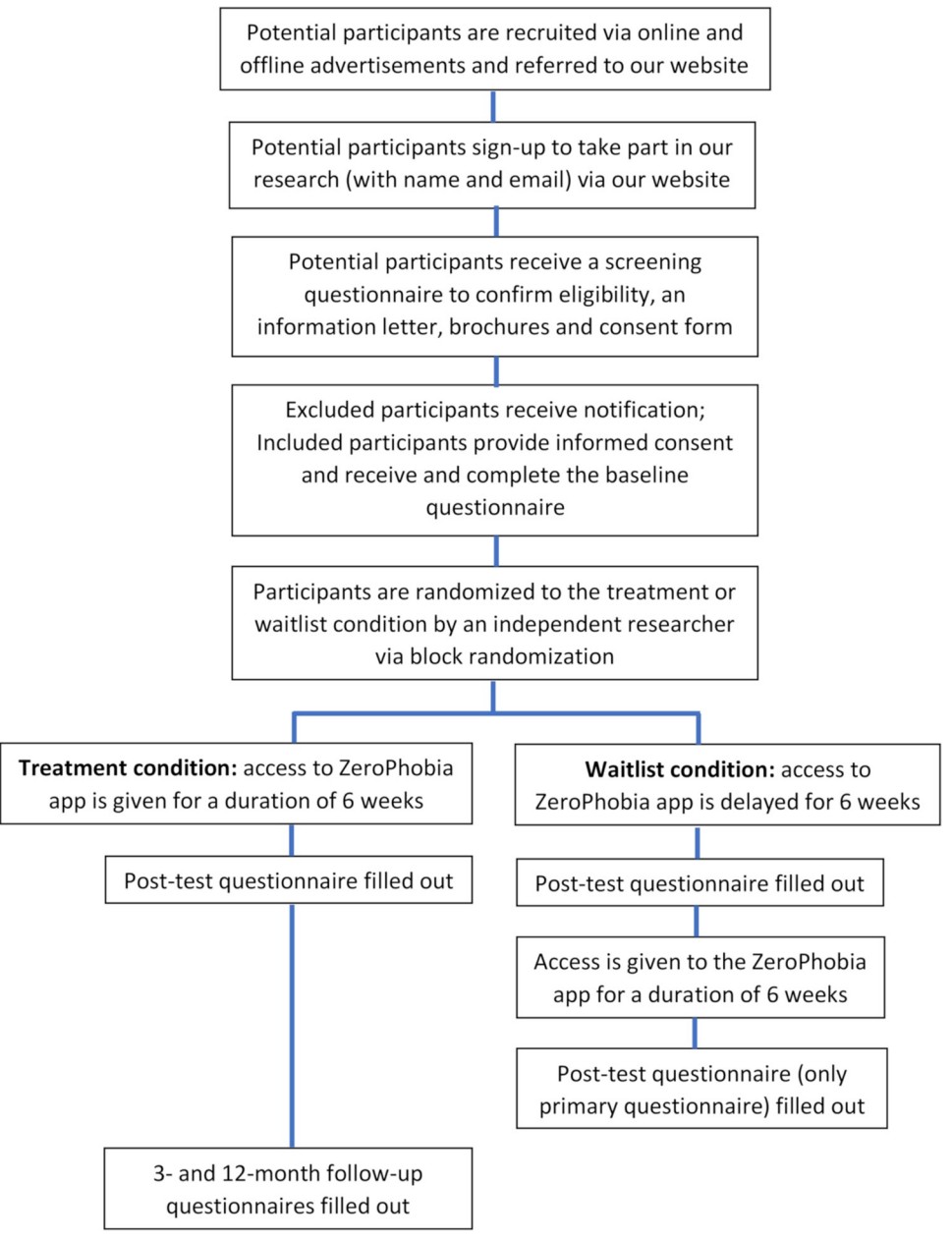

**Fig 2. Participant flow chart.**

participate. Individuals are excluded from participation when they are currently receiving other psychological treatments for arachnophobia. In addition, participants who have started their psychotropic medication within three months prior to the time of recruitment, or those who are planning to start or make adjustments to their psychotropic medication during their participation in the study will also be excluded.

## Sample size

To calculate the power of our study (with G*Power 3.1.9.7 [29]) we used our primary outcome measure of arachnophobia, the FSQ. An effect size of Cohen's $d$ = 1.11 / 1.12 was found from a

meta-analysis of VR treatments for different phobias, including fear of spiders [11, 12]. AR exposure therapy is still under-researched, but Botella and colleagues [24] explored small animal phobia using AR and found an effect size of $d = 0.80$ and the recent study by Zimmer and colleagues [26] found an effect size of $d = 0.57$. Because ZeroPhobia: Arachnophobia is both self-guided as well as outside of researcher overview, we have taken a similar effect size of $d = 0.60$. To demonstrate a difference between the treatment and waitlist control condition with a standardized effect size of $d = 0.60$ (tested on both sides), an alpha of 0.05 and a statistical power (1-beta) of 0.80, we require 45 respondents in each condition (90 respondents in total). Taking into account a dropout rate of 20%, a total of 112 respondents ($n = 56$ per condition) is required.

### Randomization, blinding, and treatment allocation

For randomization of participants we will use the random allocation software Sealed Envelope™ using block randomization with 1:1 allocation and random block sizes of 6, 8, 10, and 12. Randomization is conducted by an independent researcher. This researcher will generate a randomization list and reveal the next randomization outcome after every inclusion to the research team. This will determine whether participants are assigned to the treatment or waitlist control group. Blinding of participants and research team to the assigned condition of the participant is not possible due to the set-up of this study. However, all measurements are completed online by the participants without research team members being present.

### Procedure

Participants will be recruited from the Dutch general population through both online (such as Facebook, Instagram, and patient support webpages) and offline (national radio, newspaper/magazine articles, flyers) advertisements. Via these advertisements, interested individuals are directed to the study website, where more detailed information is available. There they can fill in a brief online response form to express interest in participation.

To determine eligibility, the research team will then provide additional information and an online screening questionnaire. Individuals who are deemed ineligible after filling in the screening questionnaire will automatically receive an email response explaining the reason why they cannot participate in the study. Potential participants will receive two email reminders (with 1-week intervals in between) to fill in this screener. If there is no response or a participant indicates they are no longer interested, these individuals will not be contacted again.

Individuals who are eligible for participation will receive an informed consent form and a participant information letter, either digitally via email or (only if specifically requested, due to COVID-19) via regular mail, with a pre-addressed return envelope enclosed. Again, participants will receive two weekly reminders (first by e-mail, then by phone) to return their signed informed consent form. After receiving their written informed consent via (e-)mail, we will send participants the online baseline questionnaire. After completing this questionnaire, participants will be randomly assigned to either the treatment or the waitlist control condition by an independent researcher. Participants will be informed of their condition and those randomized to treatment will receive instructions on how to download and install the ZeroPhobia app with a unique unlock code to access the app, and those randomized to waitlist will be told they will be contacted again in six weeks (after which they will receive and fill in the post-test questionnaire and then subsequently gain access to the ZeroPhobia app).

After downloading the app, participants are free to follow ZeroPhobia at their own pace and in their own time. Weekly emails are sent to remind and encourage participants to continue using ZeroPhobia during the six weeks (regardless of how much or how little a

participant has been using the app). Participants can repeat modules if they wish to. In the final module, participants will be encouraged to engage in real world exposures and, for instance, search for spiders outside or around their house. After six weeks, participants belonging to both conditions will receive a post-test questionnaire to fill in.

Once post-test questionnaires are complete, participants in the treatment condition are informed that they are free to continue using the ZeroPhobia app if they want to and that they will be contacted again after three and 12 months for follow-up assessments. Participants in the waitlist condition will gain access to the ZeroPhobia app and are invited to start. Similar to the treatment condition, they will also receive weekly reminders. After six weeks, participants in the waitlist condition are asked to complete one final questionnaire consisting of only the primary outcome measure.

## Intervention

ZeroPhobia: Arachnophobia is an automated self-guided app-based smartphone intervention for spider phobia, which is based on Cognitive Behavioral Therapy (CBT) protocols (e.g., [30–32]). It consists of six modules (see Table 1) that can be followed at the user's own pace. An animated and scripted virtual therapist provides educational and usage information. Each module takes between five to twenty minutes to complete, depending on how fast the user would like to progress. CBT content is provided via the virtual therapist using simple, 2D animations and voiceovers.

Module 1 provides psychoeducation including a description of what arachnophobia is and facts about spiders, such as how they behave and how harmless most of them actually are. Additionally, participants are introduced to a fictional ZeroPhobia user named Anna who shares her experiences and how she overcame her phobia.

Module 2 teaches participants about the anxiety curve and how to set realistic goals related to overcoming their fear, such as "I can work in the garden" or "I can go to the attic to take things I need".

Module 3 explains how to effectively and safely use the ARET. Importantly, after completing this module, the first AR level unlocks. The AR levels are independent from the modules. Users are instructed to sit at a table. During these AR exposures, users can choose from several options. They are first able to choose a type of spider. The list of AR spiders is presorted in order of what the researchers and a panel of test-users considered least scary to scariest. The

**Table 1.  Module overview.**

| Module | Learning objective | Additional Information |
| --- | --- | --- |
| Module 1: Achtergrond/ Background | Psychoeducation about spider phobia and how it develops | Participants are introduced to an animated fictional ZeroPhobia user, "Anna", who will provide examples of spider fear and share her experiences. |
| Module 2: Je angst te lijf /Facing your fears | Learn about the anxiety curve and how to create personal, realistic goals | Participants create personal goals, such as "I would like to be able to walk my dog through the woods without feeling anxious". |
| Module 3: Exposure | Learn about exposure and the different AR levels | After completing this module, the first AR level is unlocked. In order to unlock further AR levels, participants must report low levels of anxiety (a score of $\leq 3$ on a 1–10 scale) for the current level. |
| Module 4: Rampgedachten/ Catastrophic thoughts | Learn about automatic, catastrophic thoughts and how to identify such thoughts within themselves | Participants learn about the role of automatic, catastrophic thoughts in increasing or maintaining anxiety. Additionally, they are encouraged to reflect on how realistic their own thoughts are. |
| Module 5: Helpende gedachten/Helping Thoughts | Formulate helping thoughts to counteract the catastrophic thoughts | Participants come up with reasons for why their catastrophic thoughts are not realistic as well as think of helping thoughts that are more realistic |
| Module 6: De volgende stap/ The next step | Become inspired to begin practicing exposure in real life and formulate a fear hierarchy | The fictional character Anna helps to create an example fear hierarchy and provides recommendations on how to reward oneself for completing the steps on the hierarchy. |

options are (in ascending order of scariness): Daddy Longlegs, Cross Spider, Wolf Spider, Barn Funnel Waver, and Tarantula. Additionally, participants will be able to select a challenge level. Level 1 contains a single spider that only moves if the user taps on the screen to indicate the spider to come closer. Directly after finishing an AR level, participants are prompted to rate their anxiety ('How high was your fear at its highest during this level?') and disgust ('How much disgust did you experience during this level?') on a scale from 1 to 10. To unlock the next level, participants must rate the peak of their fear $\leq 3$. If they rate above a three, the app will prompt them to try the level again.

In Level 2 the (single) spider moves autonomously. In Level 3, there are three spiders presented at the same time who do not move unless the participant taps on the screen. In Level 4, there are again three spiders, but now they are moving autonomously. In Level 5, the user places their hand onto a table and the selected spider moves around it. Finally, in Level 6, the participant places their hand on an indicated area on the table and the selected spider walks on their hand. Participants are encouraged to practice ten minutes per day with the AR levels, alongside the three final modules. AR levels can take as short as about 2–4 minutes to complete, but participants can stay in AR as long as needed to reduce their fear.

Module 4 educates participants about automatic, catastrophic thoughts ("misinterpretations") and helps to identify these within themselves. Module 5 expands on this topic by teaching participants to construct helping thoughts they can use to combat the catastrophic thoughts identified in Module 4. In the final module, Module 6, participants are encouraged to apply the knowledge they gained in the previous Modules and begin practicing with exposure in real world settings.

The app was developed in such a way that the participant is gradually exposed to spiders. First, through the animations in the modules. Second, through a co-therapist named "Charlie", an animated spider that helps the participant with practical information about how to practice in AR through short messages. Third, as a gamified element in the app, participants are to take care of their animated "pet spider" (see Fig 3), which they are asked to give a name to and to

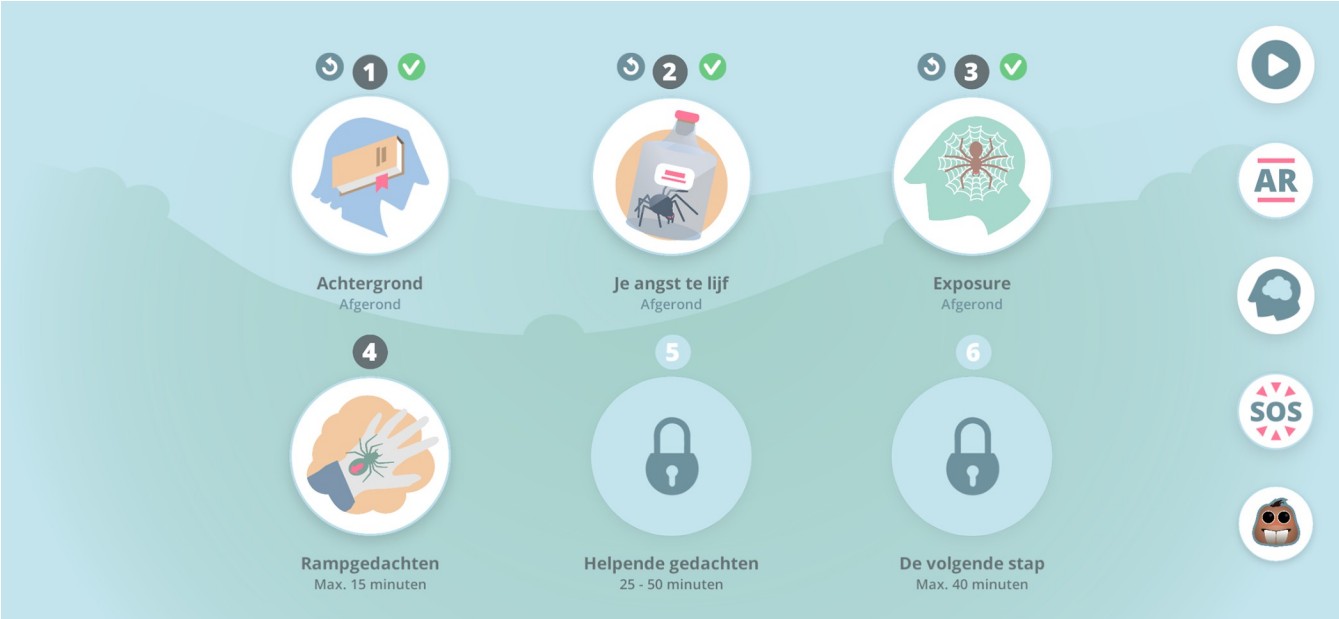

**Fig 3. Home screen (see Table 1 for translations of the module's names).**

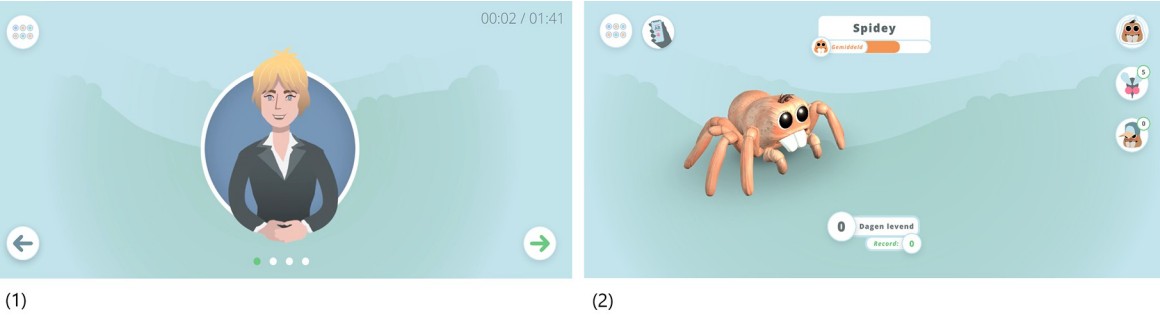

**Fig 4. Screenshots of (1) animated therapist and (2) pet spider.**

feed on a regular basis (see below). Fourth, through the different AR spiders and their movements (as mentioned before).

The app has been gamified for motivational purposes (e.g., to reduce stress; [23]). Every time a participant completes an AR exposure level, they earn a fly that can be fed to their pet spider. After five days without food, a pet spider dies. The participant can then start again with a new pet spider. After one week of exposure, irrespective of the amount of AR practice, the participant receives a prop (hat, sunglasses, shirt, shoes) to dress their pet spider. The props are lost when the pet spider dies.

Importantly, participants will continue to have access to ZeroPhobia even after the treatment phase is completed, including during follow-up. Table 1 provides an overview of all modules and Figs 3–6 contain images of the ZeroPhobia app.

## Assessments

**Primary outcome.** *The Fear of Spiders Questionnaire (FSQ; [28]).* This is an 18-item self-report questionnaire for measuring spider anxiety. An example is, *"If I saw a spider now, I would think it will harm me"*, which is rated on a 7-point Likert-scale (responses range from 1 to 7, with higher scores indicating more phobic symptoms and a maximum score of 126). The FSQ has excellent internal consistency ($\alpha$ = .94), high test-retest reliability ($r$ = .91), and is sensitive in distinguishing people with and without fear of spiders [28, 33]. Additionally, it

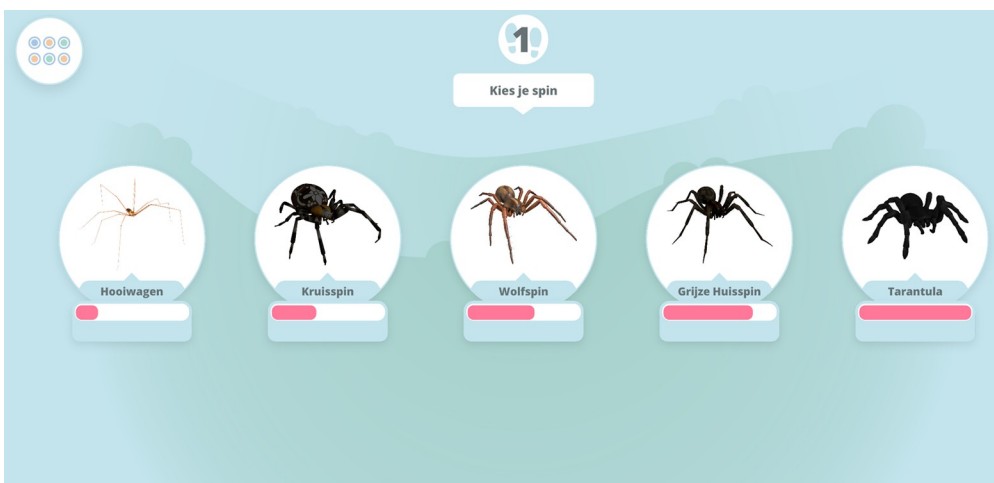

**Fig 5. Spider selection screen.**

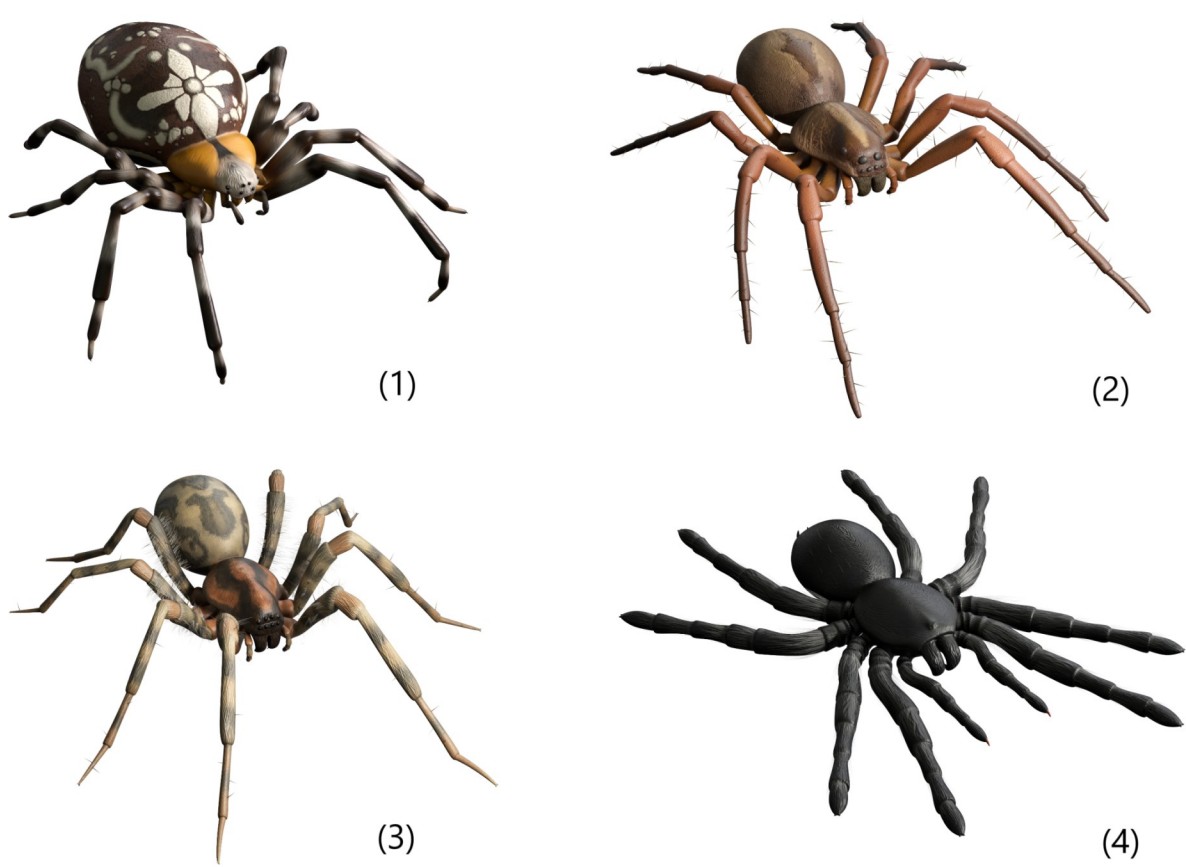

**Fig 6. AR spiders: (1) Cross spider, (2) Wolf Spider, (3) Barn Funnel Weaver, (4) Tarantula.**

demonstrated adequate convergent validity as shown by its significant positive correlation with the Spider Phobia Questionnaire ($r$ = .53; within phobic individuals).

**Secondary outcomes.** *The Spider Phobia Questionnaire (SPQ; [34]).* This self-report questionnaire is used to measure spider anxiety and consists of 31 statements that can be rated as true or false. An example is *"I avoid going to parks or on camping trips because there may be spiders around"*. Every statement that is rated as "true" will get 1 point. Therefore, the total score may range between 0–31. The SPQ has been found to be high in internal consistency [35].

*Beck Anxiety Inventory (BAI; [36]).* This self-report questionnaire consists of 21-items scores on a 4-point Likert scale ranging from 0 (not at all) to 3 (severely, I could barely stand it). It assesses to what extent participants experienced 21 common anxiety symptoms during the past week. A total possible score ranges from 0 to 63, where 0–7 is considered minimal anxiety; 8–15 is mild anxiety; 16–25 is moderate anxiety; and 26–63 is severe anxiety. Internal consistency is high (0.94), and test-retest reliability (0.73) and validity are good [37].

*Patient Health Questionnaire (PHQ-9; [38]).* This 9-item assessment will be used to measure depressive symptoms. It assesses to what degree certain problems related to depression (e.g., *"Little interest or pleasure in doing things"*) have occurred over the past 2 weeks. It is rated on a 4-point Likert scale ranging from 0 (not at all) to 3 (nearly every day), with a maximum score of 27. Major Depression is diagnosed if 5 of the 9 items are scored with a 2 or higher and 1 of these items refers to depressed mood or anhedonia. Scores of ≤ 4 suggest minimal depression [37]. This questionnaire has been found to have good sensitivity (0.71–0.84) and specificity (0.90–0.97; [39]).

*Credibility/Expectancy Questionnaire (CEQ; [40]).* The CEQ will be used at baseline to assess participants' treatment expectations. It consists of 6 items (e.g., *"At this point, how logical does ZeroPhobia seem to you?"),* rated on a 9-point Likert scale that ranges from 1 (not at all) to 9 (very; total score 54). The CEQ has been shown to have high internal consistency and good test–retest reliability [40]. The Dutch translation also was found to be valid [41]. This questionnaire will be used in combination with the Client Satisfaction Questionnaire (CSQ; see next item) to better understand whether ZeroPhobia met participants' expectations.

*Client Satisfaction Questionnaire (CSQ; [42]).* The CSQ acts as the counterpart to the CEQ, measuring satisfaction with the treatment at post-test. This questionnaire consists of 8 items, for instance regarding the quality of treatment participants received and their satisfaction with it. Items are scored on a scale from 1–4, with a total score of 32. The CSQ has been found to have good validity and internal reliability (Chronbach's alpha ranging between .83 and .93; [42]).

*System Usability Scale (SUS; [43]).* The user-friendliness of the ZeroPhobia app is measured with the SUS. It consists of 10-items rated on a 5-point Likert scale (1 = do not agree, and 5 = very much agree). Total scores are calculated and converted to range between 0–100 (see [42] for this conversion). Higher scores indicate better usability and more user-friendliness. Questions include items such as *"I think ZeroPhobia was easy to use"* and *"I had to learn a lot before I could get started with ZeroPhobia".* To pass this usability test, the app will have to be rated 70 or higher, with better products ranging into the 80s. A score of 90 or above is usually only expected for incredibly high-quality products. If the app is rated below 70, it is considered inadequate and needs to be re-assessed and improved. Reliability of the SUS is good [43].

*Igroup Presence Questionnaire (IPQ; [44]).* This 14-item questionnaire assesses realism and the feeling of "presence" or immersion in a virtual environment. For the purpose of this study, only the realism subscale (four items) was used (aligning with previous protocols for ARET investigations [18]). Each item is rated on a 7-point Likert scale ranging from -3 to 3 (total score of realism subscale is 12). Items include phrases such as, *"How real did the virtual world seem to you?".* This questionnaire has been found to be reliable (Chronbach's alpha = 0.73; [44]).

*Experience of arachnophobia.* Items were developed by the researchers in order to better understand the participants experience of their arachnophobia. This includes the item *"How long have you experienced symptoms of arachnophobia?"* with possible responses of 'less than six months,' 'six months to a year,' 'one to five years,' and 'longer than five years.' A second item (*"Have you ever had a negative experience with a spider that is related to your arachnophobia? For example, a spider that suddenly fell on you and now you are of encountering a spider again?")* will also be provided, where if answered in the affirmative, participants will then be asked to clarify what negative experience they had through a multiple-choice menu (including an open response 'other' option).

*Safety behaviors.* The use of safety behaviors is assessed by offering a multiple-choice menu of different common safety behaviors (e.g., taking or carrying medication with you, using specific items such as gloves or good luck charms, and engaging in specific rituals including checking the curtains for spiders). Participants can select as many options as they wish, and are also given the opportunity to select 'other' and report their own. A similar survey was given in relation to fear of flying (aviophobia) in previous research of ZeroPhobia [45].

*Clarification of fear.* Three questions (based on the questionnaire of Lindner and colleagues [3]) were created in order to understand what about spiders the participant fears. Response will be possible through a multiple-choice, multi-select menu, including answers such as 'I am afraid of the way spiders move' and 'I am afraid of the pain of a spider bite.' Participants are given the opportunity to also fill in their own response with an 'other' option. Next,

participants will be asked 'How much influence do the following characteristics of a spider have over the degree of anxiety you experience if you see a spider?' for eight characteristics. Characteristics include 'how big the spider is' and 'what color the spider is.' Participants can rate each characteristic on a scale from 0 to 4, where 0 means 'no influence' and 4 means 'large influence.' Finally, the third question 'Do you experience anxiety or disgust when you think of or see a spider?' will be provided. Participants can respond on a sliding scale from -2 to 2, with the options of 'only anxiety,' 'both, but more anxiety than disgust,' 'both feelings are just as strong,' 'both, but more disgust than anxiety,' and 'only disgust.'

*Ecological momentary assessment (EMA).* Participants rate their anxiety ('*How high was your anxiety at its peak during this level*?') and disgust ('*How much disgust did you experience during this level*?') on a scale from 1 to 10 directly after finishing an AR level. These "after" anxiety ratings are needed to unlock and progress to the next AR level. If a participant completes an AR level and rates their anxiety during the level above a three, they will receive pre-programmed feedback explaining that they should practice with this level again before moving forward. Additionally, these ratings will be used to increase our understanding of how efficient the AR environments are.

**Additional measures.** *Interpersonal Reactivity Index (IRI; [46]).* The IRI is a self-report questionnaire created to measure different types of empathy. The sub-section, Fantasy, will be used for the present study during baseline to understand the general ability of participants to fantasize. The IRI-Fantasy subscale consists of 7 items that are rated on a 5-point Likert scale (0 = "This does not describe me" and 4 = "This describes me very well"; total score 28). Using this subscale will give us the opportunity to investigate a person's ability to experience emotional absorption or transportation into a fantasy world. This questionnaire has good reliability and validity [46] (Dutch translation; [47]).

*Questions on professional treatment.* Single-item questions that inquire about possible other psychotherapeutic treatments that participants may engage in while in the study (either psychotherapy or the prescription and taking of psychotropic medication) will be included in the screener, but also at post-test and both follow-up time points. Therefore, if a participant does report to have started an additional treatment, we can consider this in our analyses.

## Statistical analyses

**Covariates and sample differences.** Stata 16 (StataCorp LLC) will be used to analyze the data. External validity and generalizability of the experimental sample will first be established by comparing it to the overall sample. Next, we will examine whether there are any significant differences in background characteristics between the treatment and waitlist control group to ensure successful randomization. To determine possible selective attrition, we will construct a balancing table to show how missing (outcome and input) data is divided across the two groups. In order to keep a complete experimental sample, a dummy variable will be constructed, which represents the participant's missing covariate values, which will be replaced with the means of those covariates.

**Estimating treatment effect.** We will use ordinary least squares (OLS) regression models based on intention-to-treat (ITT) to estimate the treatment effect with the main predictor being treatment assignment status. Additionally, standardizes mean differences will be calculated (Cohen's *d*). Missing outcome values for the sample will be imputed using multiple imputation exploiting pre-scores and a set of pre-specified background characteristics (gender, age, level of education, and symptom severity).

When non-random missing outcome observations are present, this can result in biased point estimates. Therefore, we will conduct a robustness analysis, which estimates the nonparametric

treatment effect bounds using the Random Forest Lee bounds procedure [48]. Finally, average treatment effects on the treated (ATT) will be estimated and consistent measures of reliable change, clinically meaningful change and numbers needed to treat will be investigated.

**Mechanisms of treatment.** To investigate underlying mechanisms that may explain observed differences, we will conduct analyses that are focused on understanding factors such as user-friendliness, treatment expectations and satisfaction, as well as usage intensity, and the level of absorption and presence. Different assumptions on the data will be used to test the robustness and sensitivity.

## Data monitoring and management

Data collection is done by researchers of the Clinical Psychology department of Vrije Universiteit Amsterdam and handled with confidentiality according to the General Data Protection Regulation. Data is collected via Survalyzer [49], except for usage data of the app (e.g., duration of using ARET or anxiety ratings during ARET). This data is sent to a secured database directly from the app using protected pathways. All data is anonymous and only marked with a four-digit identification number. Data is stored at the Vrije Universiteit Amsterdam and the identifying information is kept separate from study data, which is al password protected. Only our research team has access to this data set.

Results of the present trial will be released, regardless of whether or not these have statistical significance for our research questions, as mandated by the Dutch human research commission (CCMO). Additionally, all data that support the study findings will be made openly available through our university's data repository "Pure". Finally, a data monitoring committee is not required by the Medical Ethical Committee due to it being a low risk study, however may be audited as per the discretion of the Inspectie Gezondheiszorg en Jeugd (Healthcare and Youth Inspection, IGJ). We will not be conducting any interim analyses. Therefore, the only stopping guideline that we have is when we reach a sufficient number of participants as determined by power analysis.

## Safety considerations

Previous studies in similar samples have shown that studies with ARET can safely be carried out, without a significant risk for unwanted effects (e.g. [18, 24, 26]). Additionally, previous research on ZeroPhobia: Acrophobia demonstrated that the ZeroPhobia app can be safely carried out without serious adverse effects–e.g. no participants reported any unbearable anxiety levels during exposure exercises or any other module of the app and no other adverse events were reported [14].

Participants may discontinue their participation at any time, without providing a reason for this. In case participants experience negative side effects, such as overwhelming fear or anxiety, the research team (including an experienced clinician) is available to offer the necessary support. If needed, we will assess the situation and advise the participant to contact their primary care physician to get a referral for further psychological treatment within their vicinity.

## Protocol amendments

If we make any amendments to the protocol as it is written now, these will be included in the trial registry once Medical Ethical approval is received. Once results of this study are available they will be published in peer reviewed journals.

## Results

This study was funded on 25 September 2018. Data collection started September 2021 and we expect to finish in September 2022.

## Discussion

The aim of this study is to assess efficacy and feasibility of a new AR self-help application, ZeroPhobia: Arachnophobia, for treatment of spider phobia by using an RCT design comparing an experimental condition to a waitlist control condition. We will investigate whether 6 weeks of ZeroPhobia treatment will lead to a decrease in symptoms of spider phobia and, secondarily, whether it will decrease general anxiety and depression symptoms. We will investigate both direct treatment effects and whether these effects remain at three and 12 months follow-up. Furthermore, we investigate whether the self-help application is considered user-friendly and whether individuals feel present in the AR environments. Finally, to further understand the working mechanisms, we will investigate whether a greater reduction in spider phobia symptoms after treatment is related to higher usage intensity, higher perceived user-friendliness, more positive treatment expectations, higher treatment satisfaction, a greater ability to experience emotional absorption, and/or a higher sense of realism.

Arachnophobia is one of the most common types of specific phobia [1], which, due to the commonality of spiders, can interfere significantly with the daily life of patients who suffer from it. New technologies such as VRET and ARET may help to overcome existing accessibility burdens of current *in vivio* treatment modalities. Research into the efficacy of ARET applications for small animal phobia demonstrate promising results [20–22]. The combination of ARET delivered through a smartphone, without the guidance of a therapist, to be followed in the patients' own home environment, is an innovative and scalable method to help patients get rid of their fears. Advantages of ARET over other forms of treatment are total control over the way exposures are conducted, easy access to threatening stimuli without a risk of real danger to the patient, and lower costs than VRET [24].

The study is limited by the use of an inactive, waitlist, control group. Consequently, no direct comparisons can be made with the efficacy of current gold standard behavior treatments. Therefore, in future research, it would be of interest to compare the (cost-) effectiveness of traditional (in vivo) exposure therapy head-to-head to this new AR therapy.

In sum, the current study will use an RCT design to assess the efficacy and user-friendliness of an app-based, gamified AR treatment for arachnophobia, with the intention of making mental health care more scalable and acceptable.

## Supporting information

**S1 Checklist. SPIRIT 2013 checklist: Recommended items to address in a clinical trial protocol and related documents**∗.
(PDF)

**S1 File. Study protocol approved by ethical committee.**
(PDF)

**S2 File. WHO trial registry data set.**
(PDF)

## Acknowledgments

We would like to thank Bruno de Vos, BA, (Studio Barbaar), for designing ZeroPhobia, Doruk Eker, MSc and GianLuca di Vincenzo, MSc (Orb Amsterdam), for programming ZeroPhobia and Rufus van Baardwijk, MSc, for ZeroPhobia sound. Additionally, we would like to thank Rhiannon Sandfort for help with recruitment and participant management and Roos Rinske for help with randomization of participants.

## Author Contributions

**Conceptualization:** Marieke B. J. Toffolo, Annemieke van Straten, Jean-Louis van Gelder, Tara Donker.

**Formal analysis:** Chris P. B. J. van Klaveren, Ilja Cornelisz.

**Funding acquisition:** Tara Donker.

**Methodology:** Chris P. B. J. van Klaveren, Ilja Cornelisz.

**Project administration:** Jamie R. Fehribach.

**Software:** Jean-Louis van Gelder, Tara Donker.

**Supervision:** Marieke B. J. Toffolo, Annemieke van Straten, Tara Donker.

**Writing – original draft:** Marieke B. J. Toffolo.

**Writing – review & editing:** Jamie R. Fehribach, Chris P. B. J. van Klaveren, Ilja Cornelisz, Annemieke van Straten, Jean-Louis van Gelder, Tara Donker.

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
