## [Decision Letter · Decision Letter 0]

7 Mar 2022

PONE-D-21-38295Automated app-based augmented reality cognitive behavioural therapy for spider phobia: Study protocol for a randomized controlled trialPLOS ONE

Dear Dr. Donker,

Thank you for submitting your manuscript to PLOS ONE. After careful consideration, we feel that it has merit but does not fully meet PLOS ONE’s publication criteria as it currently stands. Therefore, we invite you to submit a revised version of the manuscript that addresses the points raised during the review process.

We look forward to receiving your revised manuscript.

Kind regards,

Walid Kamal Abdelbasset, Ph.D.

Academic Editor

PLOS ONE

Journal Requirements:

“This study has been funded by Dutch Research Council (NWO) Aspasia grant (015.014.072), NWO Creative Industrie-KIEM (KI.18.039), and an MIT R&D grant from Province of North Holland, the Netherlands.”

“Research Council (NWO) Aspasia grant (015.014.072), NWO Creative Industrie-KIEM (KI.18.039), and an MIT R&D grant from Province of North Holland, the Netherlands. All funding was awarded to TD. The funders had and will not have a role in study design, data collection and analysis, decision to publish, or preparation of the manuscript.”

“I have read the journal's policy and the authors of this manuscript have the following competing interests: TD and JLvG have developed the ZeroPhobia application, which is used in the present study in collaboration with Vrije Universiteit. ZeroPhobia is intended for commercial release. Hence, TD and JLvG will not be involved in data analysis or with any decisions related to the publication of findings.

The other authors have declared that no competing interests exist.”

5. Please upload a new copy of figures 1, 2, 3 as the detail is not clear. Please follow the link for more information: https://blogs.plos.org/plos/2019/06/looking-good-tips-for-creating-your-plos-figures-graphics/" https://blogs.plos.org/plos/2019/06/looking-good-tips-for-creating-your-plos-figures-graphics/

6.  PLOS requires an ORCID iD for the corresponding author in Editorial Manager on papers submitted after December 6th, 2016. Please ensure that you have an ORCID iD and that it is validated in Editorial Manager. To do this, go to ‘Update my Information’ (in the upper left-hand corner of the main menu), and click on the Fetch/Validate link next to the ORCID field. This will take you to the ORCID site and allow you to create a new iD or authenticate a pre-existing iD in Editorial Manager. Please see the following video for instructions on linking an ORCID iD to your Editorial Manager account: https://www.youtube.com/watch?v=_xcclfuvtxQ.

Reviewers' comments:

Reviewer's Responses to Questions

**Comments to the Author**

1. Does the manuscript provide a valid rationale for the proposed study, with clearly identified and justified research questions?

Reviewer #1: Partly

Reviewer #2: Yes

2. Is the protocol technically sound and planned in a manner that will lead to a meaningful outcome and allow testing the stated hypotheses?

Reviewer #1: Partly

Reviewer #2: Yes

3. Is the methodology feasible and described in sufficient detail to allow the work to be replicable?

Reviewer #1: Yes

Reviewer #2: Yes

4. Have the authors described where all data underlying the findings will be made available when the study is complete?

Reviewer #1: Yes

Reviewer #2: Yes

5. Is the manuscript presented in an intelligible fashion and written in standard English?

Reviewer #1: Yes

Reviewer #2: Yes

6. Review Comments to the Author

You may also provide optional suggestions and comments to authors that they might find helpful in planning their study.

Reviewer #1: Dear Authors, this protocol is very interesting and has a novel idea that might address an important public health concern. Please find some comments, below. I would love to read the revised protocol, and am also enthusiastic to read the finding of this study.

1. Abstract part, I suggest avoiding using abbreviations in the abstract part.

2. Introduction part, it will be good if the authors, clearly state what merits ARET will have over VRET?

3. Exclusion part, it is stated as to “when they are starting, adjusting, or planning to start or adjust their psychotropic medication within three months prior to the time of recruitment until the planned end of the study”. How the participants know like adjusting psychotropic medication because this is the role of the physician.

4. Sample size part, it is stated that a small study on ARET reveals that as it has 0.8 effect size, while, authors prefer to use effect size of 0.6 to calculate the sample size. But, if the authors want to have a conservative effect size, why not another effect size that yields a more conservative effect size? Or if the authors are interested to increase sample size, why not other ways of escalating the number of participants?

5. I think it will be good if the author does power analysis and add this to the protocol?

6. Procedure part, in the protocol, it is stated as “Potential participants will receive two email reminders to fill in this screener. If there is no response or a participant indicates they are no longer interested, these individuals will not be contacted again”. So, what is the time interval between two email reminders?

7. It is stated as “participants can repeat the module if they want to”. However, if some participants repeat the module, while if the others do not repeat the module, then, I think it will affect the outcomes and makes it difficult to reach conclusion. So, how do the authors deal with issues?

8. For the waiting list group, there is no placebo given for 6 weeks. How did the author see these issues?

9. Participants in the waiting list group might have exposure to spiders through other means like on the internet by themselves during 6 weeks period of study. So, how do researchers handle this?

10. I suggest adding validity, reliability, sensitivity, and specificity for FSQ?

11. Why does the author describe the outcomes that are measured by SPQ as secondary outcomes? Since the study deals with the reduction of phobia.

12. In protocol stated, “In case participants experience negative side effects, such as overwhelming fear or anxiety, the research team (including an experienced clinician) is available to offer the necessary support”. I suggest adding some information on how the participant’s help sought will be handled in the protocol?

13. I could not find the details of Module 1 to 6. So, if possible, I suggest to add this module.

Reviewer #2: The protocol seemed to have sound methodology and procedures were described in details. Furthermore, opinion from a better expert in this field would be better.

7. PLOS authors have the option to publish the peer review history of their article (what does this mean?). If published, this will include your full peer review and any attached files.

Reviewer #1: No

Reviewer #2: **Yes: **Dr. Panchanan Acharjee

---

## [Author Response · Author response to Decision Letter 0]

30 May 2022

Response to Journal Requirements:

We have checked PLOS ONE’s style requirements, and we believe our manuscript now meets these. Additionally, we have checked the manuscript for UK/US spelling and applied US spelling throughout. 

Since this is a protocol paper, there is no full/minimal data set yet that underlies the results. We therefore are unable to comply with this requirement at this moment. However, once the data of the final study is finalized we will make this publicly available through our University’s Data repository. Therefore, we have updated our Data Availability Statement as follows “Once the proposed study is finalized, we will make the data that support the findings of this study openly available in our University’s data repository “Pure”, which can then be accessed through research.vu.nl.” Please let us know if this is sufficient.

“This study has been funded by Dutch Research Council (NWO) Aspasia grant (015.014.072), NWO Creative Industrie-KIEM (KI.18.039), and an MIT R&D grant from Province of North Holland, the Netherlands.”

“Research Council (NWO) Aspasia grant (015.014.072), NWO Creative Industrie-KIEM (KI.18.039), and an MIT R&D grant from Province of North Holland, the Netherlands. All funding was awarded to TD. The funders had and will not have a role in study design, data collection and analysis, decision to publish, or preparation of the manuscript.”

Thank you for pointing this out. We have removed the sentence about financial compensation to Studio Barbaar, Orb Amsterdam and Rufus van Baardwijk from the Acknowledgements Section. Since these parties are not involved in the actual study or manuscript we did not add this information to the Funding Statement. Therefore, the funding statement does not have to be amended. 

“I have read the journal's policy and the authors of this manuscript have the following competing interests: TD and JLvG have developed the ZeroPhobia application, which is used in the present study in collaboration with Vrije Universiteit. ZeroPhobia is intended for commercial release. Hence, TD and JLvG will not be involved in data analysis or with any decisions related to the publication of findings.

The other authors have declared that no competing interests exist.”

The updated Competing Interests Statement is: 

“I have read the journal's policy and the authors of this manuscript have the following competing interests: TD and JLvG have developed the ZeroPhobia application, which is used in the present study in collaboration with Vrije Universiteit. ZeroPhobia is intended for commercial release. Hence, TD and JLvG will not be involved in data analysis or with any decisions related to the publication of findings. This does not alter our adherence to PLOS ONE policies on sharing data and materials. The other authors have declared that no competing interests exist.”

5. Please upload a new copy of figures 1, 2, 3 as the detail is not clear. Please follow the link for more information: https://blogs.plos.org/plos/2019/06/looking-good-tips-for-creating-your-plos-figures-graphics/" https://blogs.plos.org/plos/2019/06/looking-good-tips-for-creating-your-plos-figures-graphics/

We have uploaded new copies of all figures (1-6) and ensured they meet PLOS guidelines through PACE digital diagnostic tool. 

6. PLOS requires an ORCID iD for the corresponding author in Editorial Manager on papers submitted after December 6th, 2016. Please ensure that you have an ORCID iD and that it is validated in Editorial Manager. To do this, go to ‘Update my Information’ (in the upper left-hand corner of the main menu), and click on the Fetch/Validate link next to the ORCID field. This will take you to the ORCID site and allow you to create a new iD or authenticate a pre-existing iD in Editorial Manager. Please see the following video for instructions on linking an ORCID iD to your Editorial Manager account: https://www.youtube.com/watch?v=_xcclfuvtxQ.

The ORCID iD for the corresponding author has been added. 

We have reviewed the reference list and can ensure that it is complete and correct. No papers that were cited have been retracted. Only one paper has been added to the reference list; reference 29 was added in response to question 5 of reviewer 1.

Response to reviewers' comments:

Reviewer #1: Dear Authors, this protocol is very interesting and has a novel idea that might address an important public health concern. Please find some comments, below. I would love to read the revised protocol, and am also enthusiastic to read the finding of this study.

1. Abstract part, I suggest avoiding using abbreviations in the abstract part.

We thank the reviewer for this suggestion. We have removed all abbreviations from the abstract.

2. Introduction part, it will be good if the authors, clearly state what merits ARET will have over VRET?

We have made this even more explicit on page 6, line 101-107 (in marked-up manuscript): “The latter makes AR technology more easily accessible than VR for most people, because the majority of the population has access to this kind of technology (smartphone penetration rate within Western Europe and the U.S. is ≥ 75%; [19]). Additionally, when using a handheld device there is no need for an additional headset as is the case with VR, which makes the use of AR less costly. Moreover, ecological validity is arguably higher in AR, because the feared stimulus is embedded in the real environment of the individual.”

3. Exclusion part, it is stated as to “when they are starting, adjusting, or planning to start or adjust their psychotropic medication within three months prior to the time of recruitment until the planned end of the study”. How the participants know like adjusting psychotropic medication because this is the role of the physician.

It is indeed true that the physician most often determines whether a patient’s psychotropic medication has to be adjusted. However, the patient will still be aware of any adjustments if they occur. Therefore, if a participant is on a stable dose of psychotropic medication for at least 3 months prior they can start with the study, but we ask them to inform us if there were any adjustments to their medication (whether initiated by their physician or by themselves). If that’s the case we will exclude them from the analyses, since the medication change may impact the results. 

For clarification purposes, we have rephrased the sentence in the revised manuscript (page 11, line 207-211): 

“In addition, participants who have started their psychotropic medication within three months prior to the time of recruitment, or those who are planning to start or make adjustments to their psychotropic medication during their participation in the study will also be excluded.

4. Sample size part, it is stated that a small study on ARET reveals that as it has 0.8 effect size, while, authors prefer to use effect size of 0.6 to calculate the sample size. But, if the authors want to have a conservative effect size, why not another effect size that yields a more conservative effect size? Or if the authors are interested to increase sample size, why not other ways of escalating the number of participants?

We are happy to clarify this. We chose a more conservative effect size, because (as described) our application is both self-guided and without interference of a therapist or researcher. Therefore, we have less control over the actual practice with exposure therapy and thus expect a smaller effect size than studies that did involve contact with a therapist. Additionally, we reviewed the literature again and the most recent study that was published using a similar self-guided AR application (Zimmer et al, 2021) indeed found an effect size of Cohen’s d = 0.57. Therefore, we believe that our power calculation is accurate (see page 11, line 214-225). However, we recently finalized another ZeroPhobia study for fear of flying and these results demonstrated a 20% drop-out rate (results not published yet). Therefore, we believe this would be a more accurate expectation than a 40% drop-out rate in the present study. Thus, taking into account a dropout rate of 20%, a total of 112 respondents (n = 56 per condition) is required. This was adapted throughout the manuscript.

5. I think it will be good if the author does power analysis and add this to the protocol?

A power analysis was indeed already included in the protocol (page 11, line 214-225). However, to clarify we did add the program that was used, G*Power 3.1.9.7 (Faul et al., 2009) and added information about the effect size that was chosen (as discussed in the response to question 4 above).

6. Procedure part, in the protocol, it is stated as “Potential participants will receive two email reminders to fill in this screener. If there is no response or a participant indicates they are no longer interested, these individuals will not be contacted again”. So, what is the time interval between two email reminders?

Thank you for this question. There is a 1-week interval between these two reminders. We have clarified this in the procedure (page 13, line 245 and 250).

7. It is stated as “participants can repeat the module if they want to”. However, if some participants repeat the module, while if the others do not repeat the module, then, I think it will affect the outcomes and makes it difficult to reach conclusion. So, how do the authors deal with issues?

We are happy to clarify this. Although the option to repeat modules might be difficult to interpret with respect to within-treatment group comparisons, it does leave the internal validity of the between-group comparison across experimental groups unaffected. Thus, the main objective of the study (i.e., deriving an unbiased estimate for the effectiveness of using ZeroPhobia in a self-guided automatic delivery setting vs wait-list control) is not jeopardized by allowing participants in the treatment group to repeat modules. In fact, this kind of differential usage behavior is exactly what you would also expect from users in a real-life, non-RCT setting, and therefore only adds to the validity of the study. 

8. For the waiting list group, there is no placebo given for 6 weeks. How did the author see these issues?

Thank you for your question. The use of a waitlist control group is a commonly used control group in psychological RCT research, which allows for a comparison between receiving treatment and not receiving treatment – basically controlling for time passed. During the 6 weeks of trial participation, participants will not receive a placebo, because it is believed that it is unethical to knowingly provide participants with a psychological treatment that we know does not work. Therefore, participants in the waitlist condition do gain access to the app after trial participation, and therefore can still benefit from participating in the study. This is similar to other studies that were recently published (e.g., Zimmer et al., 2021). However, since the waitlist condition is an inactive control group, it does not control for non-specific factors of psychotherapy, such as the quality of the therapeutic alliance. Importantly though, since our study only uses a self-guided mobile phone application without active therapeutic guidance, such alliance does not apply. Therefore, we believe that the use of a waitlist control group is a sound set-up for our study. 

9. Participants in the waiting list group might have exposure to spiders through other means like on the internet by themselves during 6 weeks period of study. So, how do researchers handle this?

This is indeed a possibility, but the possibility of this accidental exposure to spiders during the trial is equal in the treatment condition. Therefore, we believe that by randomization we can rule out the effect of this on the treatment outcome.

10. I suggest adding validity, reliability, sensitivity, and specificity for FSQ?

As by your suggestion we have added some information regarding the internal consistency, test retest reliability and validity of the FSQ (page 17, line 345-349): “The FSQ has excellent internal consistency (α = .94), high test-retest reliability (r = .91), and is sensitive in distinguishing people with and without fear of spiders [28], [32]. Additionally, it demonstrated adequate convergent validity as shown by its significant positive correlation with the Spider Phobia Questionnaire (r = .53; within phobic individuals).”

11. Why does the author describe the outcomes that are measured by SPQ as secondary outcomes? Since the study deals with the reduction of phobia.

There can only be one primary outcome measure in a study. For determining robustness of results, we have included another spider phobia measure as a secondary outcome. 

12. In protocol stated, “In case participants experience negative side effects, such as overwhelming fear or anxiety, the research team (including an experienced clinician) is available to offer the necessary support”. I suggest adding some information on how the participant’s help sought will be handled in the protocol?

Thank you for this suggestion. In case of overwhelming fear or anxiety we will advise the participant to contact their primary care physician to get a referral for further psychological treatment within their vicinity. We have added this information on page 25, line 507-508: “If needed, we will assess the situation and advise the participant to contact their primary care physician to get a referral for further psychological treatment within their vicinity.”

13. I could not find the details of Module 1 to 6. So, if possible, I suggest to add this module.

All modules are explained in the “Intervention” section within the Methods and in Table 1 “Module overview”. Please let us know when this information is insufficient. 

Reviewer #2: The protocol seemed to have sound methodology and procedures were described in details. Furthermore, opinion from a better expert in this field would be better.

Thank you for your positive review of our manuscript.

---

## [Decision Letter · Decision Letter 1]

27 Jun 2022

Automated app-based augmented reality cognitive behavioral therapy for spider phobia: Study protocol for a randomized controlled trial

PONE-D-21-38295R1

Dear Dr. Donker,

We’re pleased to inform you that your manuscript has been judged scientifically suitable for publication and will be formally accepted for publication once it meets all outstanding technical requirements.

Kind regards,

Walid Kamal Abdelbasset, Ph.D.

Academic Editor

PLOS ONE

Additional Editor Comments (optional):

Reviewers' comments:

Reviewer's Responses to Questions

**Comments to the Author**

1. Does the manuscript provide a valid rationale for the proposed study, with clearly identified and justified research questions?

Reviewer #1: Yes

2. Is the protocol technically sound and planned in a manner that will lead to a meaningful outcome and allow testing the stated hypotheses?

Reviewer #1: Yes

3. Is the methodology feasible and described in sufficient detail to allow the work to be replicable?

Reviewer #1: Yes

4. Have the authors described where all data underlying the findings will be made available when the study is complete?

Reviewer #1: Yes

5. Is the manuscript presented in an intelligible fashion and written in standard English?

Reviewer #1: Yes

6. Review Comments to the Author

You may also provide optional suggestions and comments to authors that they might find helpful in planning their study.

Reviewer #1: Dear Authors, thank you for the response to my comments and all my comments and concerns are well addressed.

7. PLOS authors have the option to publish the peer review history of their article (what does this mean?). If published, this will include your full peer review and any attached files.

Reviewer #1: No

---

## [Editor Report · Acceptance letter]

5 Jul 2022

PONE-D-21-38295R1 

Automated app-based Augmented Reality cognitive behavioral therapy for spider phobia: Study protocol for a randomized controlled trial 

Dear Dr. Donker:

I'm pleased to inform you that your manuscript has been deemed suitable for publication in PLOS ONE. Congratulations! Your manuscript is now with our production department. 

Kind regards, 

on behalf of

Dr. Walid Kamal Abdelbasset 

Academic Editor

PLOS ONE